# The Prognostic Significance of Neutrophil to Lymphocyte Ratio (NLR), Monocyte to Lymphocyte Ratio (MLR) and Platelet to Lymphocyte Ratio (PLR) on Long-Term Survival in Off-Pump Coronary Artery Bypass Grafting (OPCAB) Procedures

**DOI:** 10.3390/biology11010034

**Published:** 2021-12-27

**Authors:** Tomasz Urbanowicz, Anna Olasińska-Wiśniewska, Michał Michalak, Michał Rodzki, Anna Witkowska, Ewa Straburzyńska-Migaj, Bartłomiej Perek, Marek Jemielity

**Affiliations:** 1Cardiac Surgery and Transplantology Department, Poznan University of Medical Sciences, 61-848 Poznan, Poland; anna.olasinska@poczta.onet.pl (A.O.-W.); michal.rodzki@skpp.edu.pl (M.R.); anna.witkowska@skpp.edu.pl (A.W.); bperek@ump.edu.pl (B.P.); mjemielity@poczta.onet.pl (M.J.); 2Department of Computer Science and Statistics, Poznan University of Medical Sciences, 60-806 Poznan, Poland; michal@ump.edu.pl; 31st Cardiology Department, Poznan University of Medical Sciences, 61-848 Poznan, Poland; ewa.straburzynska-migaj@skpp.edu.pl

**Keywords:** off-pump coronary artery bypass grafting, neutrophil to lymphocyte ratio, monocyte to lymphocyte ratio, platelets to lymphocyte ratio

## Abstract

**Simple Summary:**

Inflammatory processes are involved in the development and progression of coronary artery disease. Environmental factors, including hyperglycaemia, hyperlipidaemia, or smoking, promote endothelial disfunction. This process generates inflammation, which further exaggerates vascular wall injury. Immune cells migrate to the endothelium after its damage and promote platelets activation. Cardiac surgery activates inflammatory response. The knowledge on the intensity of immune cells activation may enable assessment of long-term consequences, including increased morbidity and mortality. The off-pump coronary artery bypass grafting technique allows to minimize systemic inflammatory reaction; however, some extent of the response still exists and may influence surgical results. The assessment of inflammatory response has undeniable value in coronary artery disease management. Several markers have been proposed; however, indices widely available from whole blood count are most valuable in daily practice. We evaluated and proposed the use of neutrophil to lymphocyte ratio (NLR), monocyte to lymphocyte ratio (MLR), and platelets to lymphocyte ratio (PLR) in the prognosis of the long-term outcomes after off-pump coronary artery bypass grafting. Patients who present with abnormally increased values of pre-operative and post-operative NLR, MLR, and PLR should undergo meticulous follow-up controls, as they are burdened with a higher risk of death.

**Abstract:**

Background: Cardiovascular diseases, apart from commonly known risk factors, are related to inflammation. There are several simple novel markers proposed to present the relation between inflammatory reactions activation and atherosclerotic changes. They are easily available from whole blood count and include neutrophil to lymphocyte ratio (NLR), monocyte to lymphocyte ratio (MLR), and platelets to lymphocyte ratio (PLR). The RDW results were excluded from the analysis. Method and results: The study based on retrospective single-centre analysis of 682 consecutive patients (131 (19%) females and 551 (81%) males) with median age of 66 years (60–71) who underwent off-pump coronary artery bypass grafting (OPCAB) procedure. During the median 5.3 +/− 1.9 years follow-up, there was a 87% cumulative survival rate. The laboratory parameters including preoperative MLR > 0.2 (HR 2.46, 95% CI 1.33–4.55, *p* = 0.004) and postoperative NLR > 3.5 (HR 1.75, 95% CI 1.09–2.79, *p* = 0.019) were found significant for long-term mortality prediction in multivariable analysis. Conclusion: Hematological indices NLR and MLR can be regarded as significant predictors of all-cause long-term mortality after OPCAB revascularization. Multivariable analysis revealed preoperative values of MLR > 0.2 and postoperative values of NLR > 3.5 as simple, reliable factors which may be applied into clinical practice for meticulous postoperative monitoring of patients in higher risk of worse prognosis.

## 1. Introduction

Cardiovascular diseases, apart from commonly known genetic, environmental, and behavioral risk factors, are related to inflammation [1,2,3]. Inflammatory cells and signaling pathways are pathological cornerstones of atherosclerosis development and progression [4]. The immunometabolism presents an interplay between metabolic diseases and inflammatory reactions [5].

The assessment of inflammatory response has undeniable value in coronary artery disease management. There are several simple novel markers proposed to present the relation between inflammatory reactions activation and atherosclerotic changes. They are easily available from the whole blood count and include neutrophil to lymphocyte ratio (NLR), monocyte to lymphocyte ratio (MLR), and platelets to lymphocyte ratio (PLR) [6,7,8,9].

MLR was postulated as an independent risk factor for coronary artery severity including future cardiovascular events [9]. NLR was reported as independent risk factor for severity of coronary lesions and predictor of adverse clinical outcomes, especially of acute coronary syndromes [10]. PLR was found as a predictor of atherosclerosis severity in stable coronary disease [11].

Surgical interventions activate inflammatory system response and may induce a chain of reactions that have an influence on patients’ survival [12,13,14]. The utility of aforementioned ratios in surgical interventions was presented in previous studies [15,16,17,18,19,20,21].

Several studies have proved excellent long-term results of surgical revascularization for complex coronary artery disease [22,23,24,25]. The superior effect of coronary artery bypass grafting is related to arterial revascularization, despite the fact that systemic inflammatory activation triggered by cardiopulmonary bypass is present [26,27,28,29]. The off-pump coronary artery bypass grafting (OPCAB) was found to be a reliable option [29]. OPCAB technique allows to omit or minimize systemic inflammatory reaction syndrome (SIRS) related to extracorporeal circulation utility [30,31,32]. Still, however, some extension of inflammatory reaction occurs during surgical interventions and lasts for 24–48 h [33].

The aim of the study was to compare presented hematological indices and long-term mortality risk in patients undergoing off-pump coronary artery bypass grafting.

## 2. Materials and Methods

The analysis including NLR, MLR, and PLR was conducted in a registry-based study performed in as single-center analysis of 682 patients who underwent off-pump coronary artery bypass grafting (OPCAB) at an experienced academic center in Poland. All participants of the study were referred for surgical revascularization of multivessel coronary atherosclerosis and were qualified for elective coronary artery bypass grafting (CABG) by the heart team [34,35]. The trial was conducted in accordance with the principles of Good Clinical Practice and the Declaration of Helsinki and approved by the Local Ethics Committee of Medical University of Poznan (approval number: 914/21). All participants were provided written informed consent.

The exclusion criteria involved patients requiring complex procedure including either concomitant valve disease requiring surgical intervention or aortic aneurysms, patients referred for surgery with acute nonST-elevation (NSTEMI) and ST-elevation (STEMI) myocardial infarction, and end-stage kidney disease requiring hemodialysis. In addition, patients affected by active or chronic inflammatory or autoimmune diseases, and steroid therapy or presenting active or past hematological proliferative diseases or oncological history were excluded as well. Patients with incomplete preoperative medical records, including whole blood count laboratory results, were disqualified.

There were standardized definitions for collecting clinical information from electronic medical records as demographics, patients’ comorbidities, available laboratory data, applied pharmacotherapy, recorded echocardiographic data, surgical details, and in-hospital outcomes were used. Preoperative hematological indices including NLR, MLR, and PLR were determined on day 1 before surgery (baseline value). NLR was calculated as neutrophils number divided by the number of lymphocytes, MLR as the number of monocytes divided by the number of lymphocytes, and PLR as the number of platelets divided by the number of lymphocytes. Postoperative NLR, MLR, and PLR were determined on first postoperative day following surgery. The RDW (red cells distribution width) results were excluded from the analysis. The other inflammatory parameters, such as C-reactive protein (CRP) or procalcitonin, were not routinely assessed.

The numbers of neutrophils, monocytes, and platelets to lymphocytes for NLR, MLR, and PLR assessment were collected by a routine hematology analyzer (Sysmex Europe GmbH, Norderstedt, Germany). Other laboratory data including markers of myocardium injury were collected one day before and one day after procedure. Data regarding long-term mortality were collected from the Polish National Health Service database.

An experienced team of cardiac surgeons performed the surgeries. Off-pump surgical revascularization is a standard technique for complex coronary artery surgery in the institution. There was no conversion to pump surgery with extracorporeal circulation administration in the presented study group.

### 2.1. Outcomes

The long-term all-cause mortality was the primary outcome in patients with stable multivessel coronary syndrome referred for surgical revascularization in off-pump technique.

### 2.2. Statistical Analysis

In this study, continuous variables were presented as mean ± standard deviation (SD) or median with interquartile range. Normally distributed data underwent analysis using an unpaired t-test. Not normally distributed data were compared with the use of Mann–Whitney U test. Categorical variables are presented in the study as frequencies and percentages and were compared using a test for proportions. Receiver operating characteristic (ROC) curve were applied for the cut-offs values of the analyzed predictors that discriminated between individuals enrolled in the study with and without mortality endpoint. The long-term mortality predictors estimated in the study was analyzed by the Cox’s proportional hazards model. There were univariate and multivariate analyses were performed with adjustment of the estimated coefficients to the coexistence of other predictors. Stepwise, backward selection procedures were applied for the multivariate analysis. The hazard ratios (HR) and their 95% confidence intervals (CI) were used to show results. The continuous parameters were transformed into binary parameters (using ROC analysis). 

## 3. Results

The study group comprised of 682 consecutive patients (131 (19%) females and 551 (81%) males) with median age of 66 (60–71) who underwent OPCAB procedures between January 2014 and December 2018 in our hospital. The co-morbidities in the studied population included arterial hypertension in 530 patients (78%), diabetes mellitus in 232 (34%), hypercholesterolemia in 396 (58%), chronic obstructive pulmonary disease (COPD) in 63 (9%), and chronic kidney disease defined as glomerular filtration rate (GFR) ≤ 60 mL/min/1.63 m^2^ according to the Cockcroft–Gault equation in 38 (6%) subjects.

The indication for surgery included: left main stem stenosis in 259 (38%) patients, followed by two vessels disease in 218 (32%) and three vessels disease in 205 (30%) patients. The mean surgery (skin-to-skin) time was 2.4 ± 0.4 h, and the mean number of performed anastomoses was 2.3 ± 0.2. None of the surgeries were performed as a redo surgery. The re-thoracotomy incidence was 21 (3%), with median time of ICU stay 25 (17–34) hours. There were only 4 cases (0.5%) of postoperative stroke and 4 cases (0.5%) of respiratory insufficiency. 

The 30-day mortality rate included no intraoperative episodes and reached the overall value of 1% (seven patients). During the median 5.3 +/− 1.9 years follow-up, there was a 87% cumulative survival rate. Table 1 present the clinical characteristics of patients enrolled into the study with division depending on mortality endpoint.

Mann–Whitney tests revealed significant differences between the presented groups, including age (*p* = 0.021), concomitant diseases including chronic obstructive pulmonary disease (*p* < 0.001), peripheral artery disease (PAD) (*p* < 0.001), and results from laboratory tests including neutrophil to lymphocyte ratio (NLR) (*p* = 0.039), monocytes (*p* = 0.022), and monocyte to lymphocyte ratio (MLR) (*p* < 0.001).

Postoperative characteristics analysis (Table 2) showed significant differences in Mann–Whitney tests including laboratory tests—lymphocytes (*p* = 0.018), neutrophils (*p* = 0.003), NLR (*p* < 0.001), MLR (*p* = 0.023), mean corpuscular hemoglobin concentration (MCHC) (*p* = 0.006). Among echocardiographic features, left ventricle diastolic diameter (*p* < 0.001) and postoperative left ventricle ejection fraction (*p* < 0.0001) were significantly differed.

We performed whole group analysis to estimate cut-off values for long-term survival rates. The factors influencing long-term mortality rate included age over 62 (*p* = 0.009), COPD (*p* < 0.001), peripheral artery disease (*p* < 0.001). Among echocardiographic parameters, preoperative left ventricle ejection fraction below 50% (*p* < 0.001) and postoperative LVEF values below 45% (*p* < 0.001) were significantly related to increased mortality rates. Among hematological indices, preoperative NLR above 2.5 (*p* = 0.019) and postoperative NLR values above 3.5 (*p* < 0.0001) were related to long-term prognosis, respectively. MLR values above 0.2 (*p* = 0.006) and 0.49 (*p* = 0.009) in preoperative and postoperative results were also related to long-term prognosis, respectively. There were two more simple parameters found to be statistically significant for long-term results, including MCHC above 21.2 (*p* < 0.001) and PLR above 136 (*p* = 0.039), as presented in Table 3.

### 3.1. Univariable Analysis

In univariate analysis, age above 62 years (HR = 1.97, 95% CI 1.19–3.24, *p* = 0.006) and comorbidities including COPD (HR = 2.77, 95% CI 1.64–4.68, *p* = 0.000) and PAD (HR = 2.36, 95% CI 1.47–3.79, *p* = 0.000) were statistically significant. Preoperative laboratory parameters presenting significant relation to long-term prognosis included NLR > 2.5 (HR = 1.75, 95% CI 1.09–2.79, *p* = 0.019), MLR > 0.2 (HR = 2.46, 95% CI 1.33–4.55, *p* = 0.004), and PLR (HR = 1.01, 95% CI 1.00–1.01, *p* = 0.039). Postoperative laboratory parameters presenting significant relation included NLR > 3.5 (HR = 2.66, 95% CI 1.72–4.12, *p* = 0.000), MLR > 0.5 (HR = 1.78, 95% CI 1.13–2.78, *p* = 0.012) and PLR > 136 (HR = 2.18, 95% CI 1.31–3.62, *p* = 0.003). Table 4 presents the Cox univariable regression model performed to determine parameters that were calculated as a significant predictors of mortality after off-pump surgical revascularization.

### 3.2. Multivariable Analysis

Parameters significant in univariable analysis were subsequently verified in a multivariate analysis. Table 3 presents the Cox multivariable regression model for significant mortality predictors after surgical revascularization in off-pump technique. Age over 62 and COPD were significant clinical factors in a multivariable analysis. The laboratory parameters including preoperative MLR > 0.2 (HR 2.46, 95% CI 1.33–4.55, *p* = 0.004) and postoperative NLR > 3.5 (HR 1.75, 95% CI 1.09–2.79, *p* = 0.019) were found significant for long-term mortality prediction in multivariable analysis. Predictive echocardiographic parameters including pre- and post-operative LV diameters and LVEF are shown in Table 5.

### 3.3. Receiver Operator Characteristics (ROC) Analysis

Receiver operator characteristics revealed age over 62 as a long-term mortality factor (AUC = 0.578, *p* = 0.019) yielding sensitivity of 73.18% and specificity of 41.14%.

#### 3.3.1. Preoperative NLR, MLR, and PLR 

Among preoperative laboratory tests, the ROC analysis has shown that the optimal cut-off points for predicting long-term mortality after OPCAB procedure based on preoperative NLR > 2.5 (AUC = 0.057, *p* = 0.025) with highest sensitivity of 68.67% and specificity of 45.52%, as presented in Figure 1a. Similar, the ROC analysis selected the NLR > 2.5 (AUC = 0.0577, *p* = 0.012) yielding sensitivity of 84.34% and specificity of 32.49%, as presented in Figure 1b.

The ROC analysis has shown that there was no significant relation between preoperative PLR and long-term mortality rate. The cut-off points estimated by ROC analysis was for PLR above 128 (AUC = 0.555, *p* = 0.09) with sensitivity of 57.83% and specificity of 54.56%, as presented in Figure 1c. 

#### 3.3.2. Postoperative NLR, MLR, and PLR

Based on a postoperative laboratory test, the ROC analysis showed that the optimal cut-off point for predicting long-term mortality after an OPCAB procedure for NLR > 3.5 (AUC = 0.640, *p* < 0.0001) yielding sensitivity of 49.40% and specificity of 75.80%, as presented in Figure 2a. 

The ROC analysis has shown that the optimal cut-off points for predicting long-term mortality after an OPCAB procedure based on MLR >0.49 (AUC = 0.577, *p* = 0.025), giving sensitivity of 59.04% and specificity of 57.19%, as presented in Figure 2b. The ROC analysis has shown that the optimal cut-off points for predicting long-term mortality after the OPCAB procedure basing on PLR >136 (AUC = 0.593, *p* = 0.004), giving sensitivity of 53.01% and specificity of 83.02%, as presented in Figure 2c.

Basing on a postoperative laboratory test, the ROC analysis has shown that the optimal cut-off point for predicting long-term mortality after the OPCAB procedure for NLR > 3.5 (AUC = 0.640, *p* < 0.0001), yielding sensitivity of 49.40% and specificity of 75.80%, as presented in Figure 2a.

The cut-off points for MLR and PLR of >0.49 (AUC =0.577, *p* = 0.025) and >136 (AUC = 0.593, *p* = 0.004) were found, with sensitivity of 59.04% and specificity 57.19%, and sensitivity 53.01% and specificity 83.02%, respectively (Figure 2b,c).

Single inflammatory parameters represented a moderate predictive value in receiver characteristic operator analysis. Therefore, we constructed a multivariable model based on demographical and clinical parameters combined with inflammatory markers.

Multifactorial models were constructed separately for preoperative and postoperative parameters.

The preoperative multifactorial model was based on the following parameters: age above 62 years (*p* = 0.0238), left ventricle ejection fraction below 50% (*p* < 0.0001), MLR above 0.2 (*p* = 0.0186), MCHC below 21.1 (*p* = 0.008). ROC analysis for the constructed model is characterized by AUC = 0.699 (*p* < 0.001), yielding sensitivity of 65.06% and specificity of 67.18%, as presented in Figure 3.

The separate postoperative multifactorial model was composed of the following parameters: age above 62 years (*p* = 0.0177), left ventricle ejection fraction below 45% (*p* < 0.0001), left ventricle diameter above 48 mm (*p* = 0.0125), and NLR above 3.5 (*p* = 0.0003). The ROC analysis for the constructed postoperative model is characterized by (AUC = 0.747, *p* < 0.001), giving sensitivity of 77.11% and specificity of 59.83%, as presented in Figure 4.

## 4. Discussion

We have presented a detailed analysis of clinical factors and hematological indices that have prognostic value for long-term mortality prediction. The major finding of our study is the significant relation between simple inflammatory markers, such as MLR, NLR, and PLR in OPCAB patients and their mortality risk. Moreover, we also constructed separate two multifactorial models for mortality prediction composed from preoperative and postoperative parameters. The inflammatory markers, in combination with demographical (age) and echocardiographic results (including left ventricle diameter and left ventricle ejection fraction), occurred as significant predictors of long-term results in off-pump surgery. We have focused on preoperative values of indices and their results obtained in early inflammatory phase after the off-pump procedure. The multivariable analysis revealed a relation between preoperative MLR above 0.2 and postoperative NLR 3.5 as predictive factors. The cut-off point for patients’ age was estimated at 62 years regarding long-term mortality. The echocardiographic parameters representing significant predictive factors included left ventricle diameter and left ventricles ejection fraction.

Several possible mechanisms may explain the relationship between inflammatory indices and cardiovascular risk. Chronic exposure to metabolic risk factors, including hyperlipidemia, hyperglycemias, and smoking, causes endothelial disfunction, further accompanied by inflammatory response [36]. Inflammation causes vascular damage elicited by immune cells [37]. The major harm cells are neutrophils, which are specified to innate immunity-limiting pathogen dissemination and further activating adaptive immunity [38]. When activated in inflammation, neutrophils generate highly reactive oxygen species, which generally contribute to pathogen inactivation and death, but also have a deleterious effect on the vascular wall [37]. Moreover, neutrophils form extracellular traps (NETs) that help to destroy pathogens but may also promote thrombosis and coagulation [38]. After damage to the endothelium, neutrophils are recruited even before platelets and promote platelet activation and deposition [39]. Activated platelets in turn facilitate the recruitment of inflammatory cells to the atherosclerotic lesion and release plenty of mediators, thus enriching the inflammatory milieu [40]. Several other neutrophils’ mediators are also released, recruiting other leucocytes, or disrupting the endothelial cell structure [37]. Neutrophils also play a sufficient role in later stages of atherosclerotic lesion formation, aggravating and sustaining the chronic inflammatory setting and recruitment of circulating monocytes [41]. Monocytes are pro-inflammatory cells of innate immunity, producing cytokines, chemokines, and reactive oxidative species [42]. After recruitment, monocytes in turn take up oxidized lipids and form foam cells, which constitute the atherosclerotic plaque core in arterial wall [37,43]. Consecutively, monocytes’ secretion of proinflammatory cytokines (interleukin-1, TNF-alfa), enzymes (myeloperoxidase, matrix metalloproteinases), and growth factors occurs and leads to formation of atherogenic dysfunctional lipoproteins, facilitation of plaque destabilization, and development of atherothrombotic complications [36,42]. Contribution of monocytes in atherosclerosis also comprises of angiogenesis and tissue repair [42]. Progressing plaques may rupture, causing acute ischemic events and clinically unstable angina, myocardial infarction, or stroke. Endothelial wall rupture triggers activation of platelets, which leads to several processes included in thrombus formation. In patients with coronary artery disease, antiplatelet therapy is crucial in the prevention and treatment of both its stable and unstable stages.

Conversely, lymphocytes may play protective or pro-atherogenic roles, depending on the cell subset, either T-cells or B-cells or natural killer T (NK T) cells [44]. The number of B-cells is low, and T-cells high in atherosclerotic plaque [45]. In brief, Th1 cells secrete INF-γ, promote endothelial dysfunction and lipid accumulation in macrophages, thus exerting a pro-atherogenic effect, and regulatory T cells induce tolerance, inhibition of atherogenic T subsets, and suppression of inflammation [44,46]. The role of Th2 is more complex as both pro- and anti-atherogenic actions have been described [46]. B-cells produce antibodies, which may exert a protective action by inhibiting the uptake of oxidized lipids by macrophages and by favouring the clearance of apoptotic bodies, or may enhance the uptake of cholesterol favouring atherosclerosis, depending on the antibody type [44]. Therefore, the role of lymphocytes, though believed as atheroprotective, is complex. A low lymphocyte count was described to be linked to inferior cardiovascular outcomes in individuals with coronary artery disease and congestive failure of the heart [47]. 

This brief summary of the pro- or anti-atherogenic effects of different blood cells and particles points out the purpose of NLR, MLR, and PLR use in the assessment of coronary artery disease. We proved its significance in prediction of long-term survival after OPCAB. The whole blood analysis is a simple and routine test but may provide crucial prognostic information for practitioners. Patients who present with abnormally increased values of pre-operative and post-operative NLR, MLR, and PLR should undergo meticulous follow-up controls, as they are burdened with a higher risk of death.

### 4.1. Neutrophil to Lymphocyte Ratio (NLR)

Neutrophil to lymphocyte ratio is considered a useful inflammatory marker. Increased preoperative values of NLR were presented by Weedle et al. [48] to be associated with postoperative atrial fibrillation in cardiac surgery. Patients who develop atrial fibrillation after heart surgery experience acute oxidative stress as a complication related to inflammatory reaction following surgery [49]. Giakoumidakis et al. in their study found a relationship between perioperative NLR and higher mortality, both in-hospital and 30-day, and prolonged postoperative hospitalization [50]. Another relatively common complication, considered an autoimmune phenomenon, is post-pericardiotomy syndrome (PPS). Sevuk at al. in their study presented the predictive value of NLR with cut-off point of 8.34 for PPS risk. The overall mortality risk after cardiac surgery and its relation to NLR was postulated by Green at al. [51]. Their results were based on preoperative values including temporal stability of the presented ratio. We investigated preoperative NLR to predict risk for long-term survival, including preoperative and perioperative values. Afterwards we focused on perioperative changes of the aforementioned ratio as a possible marker of inflammatory reactivity secondary to surgical injury. Closer follow-up of patients undergoing surgical procedures based on perioperative NLR was postulated by Bath et al. [52]. NLR increase in early inflammatory phase after cardiac surgery procedures with concomitant activation of CD16 mean fluorescence index (MFI) (non-classical monocytes) was postulated in a previous report supporting the use of NLR as an inflammatory marker for outcomes prediction [53]. Preoperative NLR values above 3.36 were found significant for mortality prediction in a 30-day analysis by Haran et al. [54]. The relationship between preoperative values of NLR above 2.6 and long-term survival after heart surgery was presented by Silberman et al. [55]. Studies present a relationship between NLR and cardiac surgery procedures performed with cardiopulmonary bypass application. Utility of extracorporeal circulation indicate systemic inflammatory response (SIRS) [56,57]. Aldemir et al., in their report, presented significant NLR differences in the early inflammatory phase following surgical revascularization performed off-pump and on-pump [58]. The next reports presented by our group pointed out the relation between NLR results obtained in the early postoperative phase and long-term results in OPCAB patients [59,60]. NLR was investigated in acute postoperative kidney dysfunction after surgical revascularization and pointed out as a significant preoperative factor by Parlar et al. [61]. Interestingly, the preoperative NLR and risk for venous grafts occlusion was also inversely correlated [62]. The relation between elevated NLR and worse venous grafts patency rate can be explained by endothelial shear stress activation of pro-inflammatory signalling pathways [63]. Sterpetti et al. in vitro and in vivo models showed the differences in interleukins and TNF- alfa release by venous grafts’ endothelial cells in relation to hemodynamic stress. We present the results of early reperfusion, suggesting that overactivity of the inflammatory system following off-pump procedures can predispose patients for graft endothelium injury as proinflammatory cytokines are released. The role of immunomodulation in vein graft failure in humans was reviewed by Baganha et al. [64]. Immune cells in long-term intimal hyperplasia and accelerated atherosclerosis were presented on experimental and clinical basis. This is one of the possible explanations of pathophysiology of vein graft occlusion as better understanding of the processes taking place in endothelium would give a chance for better failure prevention [65]. We hypothesize that graft failure prevention should be focused on factors similar to those linked to cardiac graft vasculopathy. The relation between immune and non-immune agents are underlined including reperfusion injury, coronary endothelial dysfunction, and inflammation [66]. The meta-analysis performed by Jackson et al. on patients referred for vascular procedures link the preoperative NLR to all-cause mortality and indicate the relation between immune and non-immune individual predisposition to over reactivity of the inflammatory system [67]. The results from our study, presenting increased postoperative NLR (above 3.5) as a significant mortality factor in multivariable analysis, is a compromise with previous reports.

### 4.2. Monocytes to Lymphocyte Ratio (MLR)

The incidence of diseases with an inflammatory etiopathology has augmented, especially in societies with high socioeconomic status [68] development. The genesis of atherosclerosis-related diseases is believed to be related to reprogramming of the innate immune system as a result of activation and recruitment of monocyte-derived macrophages [69]. The activation of endothelium induces monocytes arrest onto the vessel wall. They transmigrate into the arterial wall and mature into macrophages [70]. In animal models, monocytes enter into the atherosclerotic plaques more readily if hypercholesterolemia co-exists [71]. MLR was shown to be an independent risk factor for the severity of cardiovascular diseases [72]. The atherosclerotic plaque development begins at sites of disturbed flow and subsequent macrophage accumulation [73]. The MLR was evaluated in propensity score analysis regarding four-year mortality risk following surgical revascularization with cardiopulmonary bypass application (on-pump technique) and was presented by Zhou et al. [74]. The results of our study revealed in a multivariable analysis that preoperative MLR > 0.2 was regarded as a significant factor for long-term mortality following off-pump revascularization. Preoperative MLR in multiple logistic regression analyses by Oksuz et al. were shown to be an independent predictor of venous graft failure [75]. In a retrospective observational study by Cai et al. performed on STEMI patients undergoing percutaneous procedures, MLR was related to worse overall prognosis [76]. 

### 4.3. Platelets to Lymphocyte Ratio (PLR)

PLR reflects the relationship between prothrombotic and inflammatory state. This ratio was first used in oncological studies as a marker of worse prognosis [77]. Subsequently it gathered growing interest in patients with coronary artery disease since both platelets and lymphocytes, as mentioned above, are included in inflammatory reaction. Elevated platelet count is by itself associated with worse cardiovascular outcomes [78,79] and high on-treatment platelet reactivity and resistance to antiplatelet therapy [80]. This may result from more advanced inflammatory response. In turn, a low lymphocyte count is associated with a higher risk of subsequent cardiac events after episodes of unstable angina [81]. Therefore, joint platelet and lymphocyte count should logically reflect both pathways of inflammation exaggeration and thrombosis burden in atherosclerosis. Indeed, PLR was evaluated in several studies. Yüksel et al. [82] showed that high PLR level is associated with the severity of coronary atheromatic changes. The pre-procedural PLR > 111 predicted severe atherosclerosis with sensitivity of 61% and specificity of 59%. Several subsequent studies confirmed a similar result [83,84]. A high level of PLR was also observed in instable angina with chronic total occlusion and poorly developed coronary collateral circulation, independent from platelet or lymphocyte count alone [85]. Azab et al. [86] found PLR to be an independent significant predictor of long-term mortality after non-ST elevation myocardial infarction. That finding was not solely related to lymphocytopenia as it was also true for patients with normal lymphocyte count. PLR > 176 was a significant predictor of four-year all-cause mortality after exclusion of 30-day deaths. The authors concluded that elevated PLR may be treated as a predictor of long-term mortality rather than a marker of acute syndrome. In patients with STEMI treated with coronary angioplasty, PLR was a no-reflow phenomenon predictor [87] and long-term mortality [88]. The usage of PLR, obtained before angioplasty, for adverse events and long-term mortality prediction, has been reported [89,90,91,92]. Qiu et al. [85] performed a meta-analysis of 14 studies presenting an association between PLR and chronic coronary syndrome. On the basis of current evidence, the authors pointed out the usefulness of PLR for predicting severe stenosis, collateral circulation, and coronary slow flow.

Our study group included patients with high prevalence of cardiovascular risk factors including hyperlipidemia, diabetes mellitus, and COPD. All presented diffuse coronary lesions evaluated in coronary angiography. Therefore, it represents subjects with the highest cardiovascular risk. Probably, this may explain our result of non-significant difference between groups, though the absolute median PLR values were higher in the non-survivors’ group. Also, in our analysis, post-operative lymphocyte count was higher in survivors. In the univariate analysis, PLR > 136 had a prognostic value for long-term mortality. Şaşkın et al. [93] noted a relationship between PLR and mortality but in a short postoperative period of 30 days. Indeed, there are no reports concerning the impact of PLR on long-term mortality after bypass surgery. Studies evaluating the usefulness of PLR in CABG patients reported its highest importance for prediction of acute kidney injury (AKI) [94] and post-operative atrial fibrillation [95]. Both complications are obviously associated with inflammatory response and domination of neutrophils engagement; therefore, NLR was also pointed out [96], as mentioned before. Navani et al. [97] however, in the large analysis of 1457 patients, did not confirm PLR significance for postoperative atrial fibrillation occurrence subgroups with PLR higher and lower than cut-off value of 86 derived from ROC analysis.

### 4.4. Study Limitation

Our study has some limitations and strengths. We analyzed a relatively large group of patients in the 5.3 +/− 1.7-year period. The first limitation is that it is a single center study. However, it provided us with the possibility to collect a group of patients operated on by the same experienced surgical team with the same technique, which precludes variations among different centers. Secondly, we only evaluated all-cause mortality without profound analysis of causes of the deaths. 

## 5. Conclusions

Hematological indices NLR and MLR can be regarded as significant predictors of all-cause long-term mortality after surgical revascularization performed with off-pump technique, especially in combination with demographical (age above 62 years) and echocardiographical parameters (preoperative left ventricle ejection fraction below 50% and postoperative below 45%, respectively, and left ventricle diameter postoperative diameter above 48 mm). Multivariable analysis revealed preoperative values of MLR > 0.2 and postoperative values of NLR > 3.5 as simple, reliable factors which may be applied into clinical practice for meticulous postoperative monitoring of patients with a higher risk of worse prognosis. 

## Figures and Tables

**Figure 1 biology-11-00034-f001:**
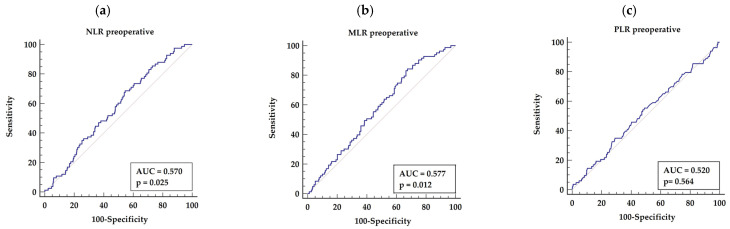
Receiver characteristics operator for long-term mortality basing on preoperative indices (NLR, MLR and PLR (**a**–**c**)). Abbreviations: AUC—area under the curve, M/L—monocyte/Lymphocyte, N/L—neutrophil/lymphocyte, P/L—platelets/lymphocyte.

**Figure 2 biology-11-00034-f002:**
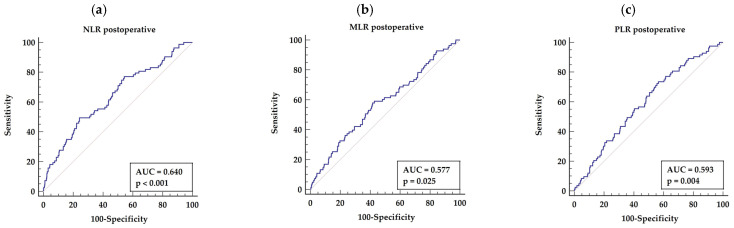
Receiver characteristics operator for long-term mortality basing on postoperative indices (NLR, MLR and PLR (**a**–**c**)). Abbreviations: AUC—area under the curve, M/L—monocyte/lymphocyte, N/L—neutrophil/lymphocyte, P/L—platelets/lymphocyte.

**Figure 3 biology-11-00034-f003:**
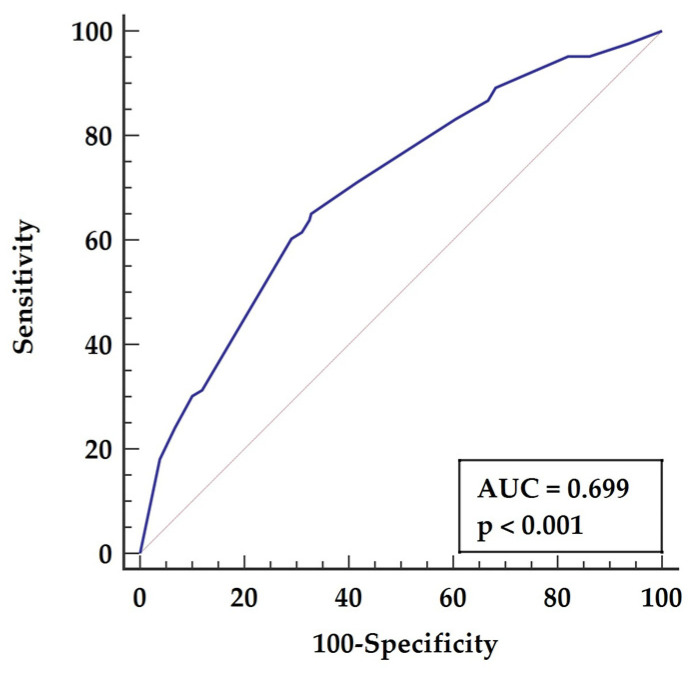
Receiver characteristics operator for long-term mortality basing on a preoperative multifactoral model. Abbreviations: AUC—area under the curve.

**Figure 4 biology-11-00034-f004:**
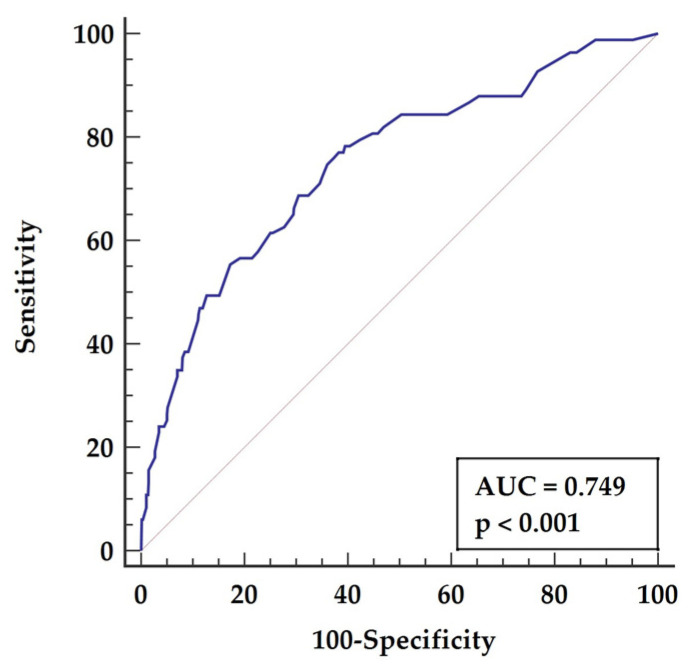
Receiver characteristics operator for long-term mortality based on postoperative multifactoral model. Abbreviations: AUC—area under the curve.

**Table 1 biology-11-00034-t001:** Demographical, clinical, and laboratory characteristics of the presented groups.

Parameters	Survivors	Deaths	*p*-Value
	No = 598 (%)	No = 84 (%)	
Demographical data			
1. Gender (M/F)	480 (80%)/118 (20%)	71 (84%)/13 (16%)	0.589
2. Age (years)	64 (59–70)	67 (62–73)	0.021 *
Comorbidities			
1. Arterial hypertension (*n* (%))	459 ((78%)	71 (85%)	0.109
2. Diabetes mellitus (*n* (%))	209 (35%)	31 (37%)	0.676
3. Hypercholesterolemia (*n* (%))	353 (59%)	43 (51%)	0.173
4. COPD (*n* (%))	45 (8%)	18 (21%)	<0.001 *
5. PAD (*n* (%))	84 (14%)	18 (21%)	<0.001 *
6. kidney failure (*n* (%))	33 (6%)	5 (6%)	0.768
Echocardiography:			
1. preoperative LV diameter (%)	48 (44–52)	50 (45–54)	0.006 *
2. preoperative LVEF (%)	55 (50–60)	50 (45–60)	<0.001 *
Preoperative laboratory tests:			
1. WBC × 10^9^/L (median (Q1–Q3))	7.7 (6.5–9.1)	7.9 (6.8–9.2)	0.657
2. Lymphocytes × 10^9^/L (median (Q1–Q3))	1.8 (1.4–2.3)	1.8 (1.4–2.0)	0.196
3. Neutrophils × 10^9^/L (median (Q1–Q3))	4.9 (4–6.2)	5.2 (4.2–6.4)	0.165
3. NLR (median (Q1–Q3))	2.7 (2–3.7)	2.9 (2.3–4.0)	0.039 *
4. Hb × 10^9^/L (median (Q1–Q3))	8.8 (8.2–9.3)	8.6 (7.9–9.2)	0.266
5. Platelets × 10^3^/uL (median (Q1–Q3))	221 (189–266)	228 (192–268)	0.559
6. Monocytes × 10^9^/L (median (Q1–Q3))	0.5 (0.4–0.6)	0.5 (0.4–0.6)	0.022 *
7. MLR (median (Q1–Q3))	0.3 (0.2–0.3)	0.3 (0.2–0.3)	<0.001 *
8. MCHC (mmol/dL) (median (Q1–Q3))	21.2 (20.7–21.6)	20.9 (20.5–21.3)	0.378
9. PLR (mean +/− SD)	134 +/− 63	148 +/− 79	0.105

COPD—chronic obstructive pulmonary disease, Hb—hemoglobin, LV—left ventricle, LVEF—left ventricle ejection fraction, MCHC—mean corpuscular hemoglobin concentration, MLR—monocyte to lymphocyte ratio, NLR—neutrophil to lymphocyte ratio, No—number, PAD—peripheral artery disease, PLR—platelets to lymphocyte ratio, SD—standard deviation, WBC—white blood cells. *—statistically significant difference.

**Table 2 biology-11-00034-t002:** Postoperative characteristics.

Parameters	Survivors	Deaths	*p*-Value
	No = 598	No = 83	
Surgical characteristics:			
1.number of grafts (*n*) (mean/SD)	2.3 +/− 0.2	2.3 +/− 0.2	0.843
2. skin-to-skin time (h) (mean/SD)	2.4 +/− 0.4	2.4 +/− 0.3	0.782
Laboratory test results:			
1. WBC × 10^9^/L (median (Q1–Q3))	8.4 (6.9–10.2)	8.8 (7.4–10.7)	0.118
2. Lymphocytes × 10^9^/L (median (Q1–Q3))	1.9 (1.5–2.5)	1.8 (1.3–2.2)	0018 *
3. Neutrophils × 10^9^/L (median (Q1–Q3))	5 (3.7–6.4)	5 (3.7–6.4)	0.003 *
3. NLR (median (Q1–Q3))	2.6 (1.8–3.5)	5.6 (4.5–7.2)	<0.001 *
4. Hb (mmol/dL) (median (Q1–Q3))	6.9 (6.5–7.3)	6.9 (6.5–7.3)	0.609
5. Platelets × 10^3^/uL (median (Q1–Q3))	278 (230–343)	290 (221–382)	0.457
6. Monocytes × 10^9^/L (median (Q1–Q3))	0.9 (0.7–1.1)	0.9 (0.7–1.2)	0.696
7. MLR (median (Q1–Q3))	0.5 (0.3–0.6)	0.5 (0.4–0.7)	0.023 *
8. MCHC (mmol/dL) (median (Q1–Q3))	21.3 (20.8–21.7)	21.1(20.7–21.6)	0.006 *
9. PLR (mean +/− SD)	134 +/− 63	186 +/− 96	0.869
10. Troponin-I (median (Q1–Q3))	1.6 (0.8–3.8)	1.8 (0.6–5.4)	0.578
Echocardiography:			
1. LV diameter (mm) (median (Q1–Q3))	47 (44–51)	50 (46–55)	<0.001 *
2. postoperative LVEF (%) (median (Q1–Q3))	60 (50–60)	45 (40–55)	<0.001 *
Overall duration of hospitalization (days) (median (Q1–Q3))	12 (7–13)	12 8–13)	*p* = 0.783

Abbreviations: Hb—hemoglobin, LV—left ventricle, LVEF—left ventricle ejection fraction, MCHC—mean corpuscular hemoglobin concentration, MLR—monocyte to lymphocyte ratio, NLR—neutrophil to lymphocyte ratio, PLR—platelets to lymphocyte ratio, SD—standard deviation, WBC—white blood cells, No—number. *—statistically significant difference.

**Table 3 biology-11-00034-t003:** Parameters related to long-term prognosis following OPCAB procedure.

Parameter	Cut-off Point	AUC	Sensitivity (%)	Specificity (%)	*p*-Value
Demographical:					
1. Age	>62 years	0.578	73.81	41.14	*p* = 0.019
Echocardiography:					
1. Preoperative LV	LV > 49 mm	0.594	51.81	63.16	<0.005
2. Preoperative LVEF	LVEF <50%	0.639	61.45	60.27	<0.001
3. Postoperative LV	LV > 48 mm	0.633	67.47	58.74	<0.001
4. Postoperative LVEF	LVEF < 45%	0.718	53.01	83.02	<0.001
Haematological indices:					
1. Preoperative NLR	NLR >2.5	0.570	68.67	45.52	0.025
2. Postoperative NLR	NLR >3.5	0.640	49.40	75.80	<0.001
3. Preoperative MLR	MLR >0.2	0.577	84.34	32.49	0.012
4. Postoperative MLR	MLR> 0.49	0.577	84.34	57.19	0.025
5. Preoperative MCHC	MCHC < 21.1	0.613	59.04	56.80	<0.001
6. Postoperative MCHC	MCHC < 21.2	0.581	65.48	60.10	0.016
7. Postoperative PLR	PLR >136	0.593	55.95	57.79	0.039

Abbreviations: AUC—area under the curve, COPD—chronic obstructive pulmonary disease, LVEF—left ventricle ejection fraction, MCHC—mean corpuscular hemoglobin concentration, MLR—monocyte to lymphocyte ratio, NLR—neutrophil to lymphocyte ratio, PAD—peripheral artery disease, PLR—platelets to lymphocyte ratio.

**Table 4 biology-11-00034-t004:** Cox regression univariable analysis.

Parameter	HR	95% CI	*p*-Value
Demographical and clinical:			
1. Age	1.04	1.01–1.07	0.006
2. Age >62 years	1.97	1.19–3.24	0.007
3. COPD	2.77	1.64–4.68	<0.001
4.PAD	2.36	1.47–3.79	<0.001
Preoperative parameters:			
1. WBC	1	0.93–1.07	0.984
2. NLR > 2.5	1.75	1.09–2.79	0.019
3. MLR > 0.2	2.46	1.33–4.55	0.004
4. MCHC	0.67	0.47–0.94	0.021
5. MCHC > 21.1		1.05–2.65	0.028
6. PLR	1.01	1.00–1.01	0.039
Postoperative parameters:			
1. WBC	1.05	1.02–1.08	0.004
2. Lymphocytes	0.67	0.47–0.94	0.02
3. Lymphocytes < 2.6	2.34	1.08–5.08	0.032
4. Neutrophils	1.12	1.07–1.16	<0.001
5. Neutrophils > 5.3	1.92	1.23–3.00	0.004
6. NLR	1.16	1.10–1.22	<0.001
7. NLR > 3.5	2.66	1.72–4.12	<0.001
8. MLR > 0.49	1.78	1.13–2.78	0.012
9. PLR	1	1.00–1.01	0.005
10. PLR > 136	2.18	1.31–3.62	0.003
Echocardiography:			
1. Preoperative LV	1.04	1.01–1.07	0.21
2. Preoperative LV > 49 mm	1.53	0.98–2.38	0.058
3. Preoperative LVEF	0.96	0.94–0.98	<0.001
4. LVEF < 50%	2.31	1.47–3.61	<0.001
5. Postoperative LV	1.06	1.03–1.09	<0.001
6. Postoperative LV > 48 mm	2.41	1.50–3.87	<0.001
7. Postoperative LVEF	0.94	0.93–0.96	<0.001
8. Postoperative LVEF < 45%	3.49	2.21–5.52	<0.001

Abbreviations: COPD—chronic obstructive pulmonary disease, Hb—hemoglobin, LV—left ventricle, LVEF—left ventricle ejection fraction, MCHC—mean corpuscular hemoglobin concentration, MLR—monocyte to lymphocyte ratio, NLR—neutrophil to lymphocyte ratio, PAD—peripheral artery disease, PLR—platelets to lymphocyte ratio, WBC—white blood cells.

**Table 5 biology-11-00034-t005:** Multivariable analysis.

Parameter	HR	95% CI	*p*-Value
Demographical and clinical:			
1. Age above 62 years	1.75	1.05–2.91	0.03
2. COPD	3.01	1.74–5.24	0
Laboratory parameters:			
1. Postoperative WBC	1.05	0.93–1.07	0.984
2. Postoperative NLR > 3.5	1.75	1.09–2.79	0.018
3. Preoperative MLR > 0.2	1.98	1.05–3.68	0.034
Echocardiography:			
1. Preoperative LV > 49 mm	0.46	0.24–0.85	0.014
2. Postoperative LV > 48 mm	2.53	1.29–4.95	0.007
3. Postoperative LVEF	0.95	0.93–0.97	0

COPD—chronic obstructive pulmonary disease, LV—left ventricle, LVEF—left ventricle ejection fraction, MCHC—mean corpuscular hemoglobin concentration, MLR—monocyte to lymphocyte ratio, NLR—neutrophil to lymphocyte ratio, WBC—white blood cells.

## Data Availability

All data will be available under correspondence e-mail address for three years following the publication after request that would be justifiable.

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
