# Peer review of "The Prognostic Significance of Neutrophil to Lymphocyte Ratio (NLR), Monocyte to Lymphocyte Ratio (MLR) and Platelet to Lymphocyte Ratio (PLR) on Long-Term Survival in Off-Pump Coronary Artery Bypass Grafting (OPCAB) Procedures"

_biology, 2021, doi:10.3390/biology11010034_

Round 1

Reviewer 1 Report

Hematological indices are not a new area of ​​research; they are widely used, among other things, to determine the prognosis. The authors obtained rather low values ​​of sensitivity and specificity. The authors did not try to consider the combination of postoperative NLR> 3.5 and preoperative MLR> 0.2 as a predictor factor. If both conditions are met simultaneously, how will the forecast change? Small remarks: 1. Key words are given as abbreviations, maybe it is better to decipher them? 2. PLR (median (Q1-Q3)) in table 1 are shown in a different form compared to the rest of the data in this table. Why? Table 2 shows a similar picture. 3. Table 3 shows the value of 22.51627 for AUC, is this a typo? The value ranges from 0 to 1. 4. Many citation errors [Error! Bookmark not defined.]. 

Author Response

           Poznan, December 18th, 2021

Dear Reviewer,

Thank you for your valuable comments. We corrected the manuscript according to them.

As follows:

The authors did not try to consider the combination of postoperative NLR> 3.5 and preoperative MLR> 0.2 as a predictor factor. If both conditions are met simultaneously, how will the forecast change?

Dear Reviewer, thank you for your valuable suggestion. The simultaneous results were still within medium sensitivity and specificity.  We focused on three most relevant inflammatory ratios related to lymphocytes (NLR, MLR, PLR) and the aim of the study was to compare their statistical strength for mortality prediction secondary to inflammatory reactions after OPCAB procedures.

We decided to construct multifactorial models and included the result in manuscript as presented below:

Single inflammatory parameters represented a moderate predictive value in the receiver characteristics operator analysis. Therefore, we constucted a mulitvariable model basing on demographical and clinical parameters combined with inflammatory markers.

The multifactorial models were constructed separately for preoperative and postoperative parameters.

The preoperative multifactorial model was based on the following parameters: age above 62 years (p=0.0238), left ventricle ejection fraction below 50% (p<0.0001), MLR above 0.2 (p=0.0186), MCHC below 21.1 (p=0.008). The ROC analysis for constructed model is characterized by AUC =0.699 (p<0.001) yielding sensitivity 65.06% and specificity 67.18% as presented in Figure 3.

Figure 3.

Receiver characteristics operator for long-term mortality basing on preoperative multifactoral model.

Abbreviations: AUC – area under the curve

The separate postoperative multifactorial model was composed of the following parameters: age above 62 years (p=0.0177), left ventricle ejection fraction below 45% (p<0.0001), left ventricle diameter above 48 mm (p=0.0125) and NLR above 3.5 (p=0.0003). The ROC analysis for constructed postoperative model is characterized by (AUC =0.747, p<0.001) giving sensitivity 77.11% and specificity 59.83% as presented in Figure 4.

Figure 4.

Receiver characteristics operator for long-term mortality basing on postoperative multifactoral model.

Abbrevations: AUC – area under the curve

Added in discusion:

Moreover, we also constructed the separate two multifactorial models for mortality prediction composed from preoperative and postoperative parameters. The inflammatory markers in combination with demographical (age) and echocardiographical results (including left ventricle diameter and left ventricle ejection fraction) occurred as significant predictors of long term results in off-pump surgery.

And in conclusion:

Hematological indices, NLR and MLR can be regarded as significant predictors of all-cause long-term mortality after off – pump surgical revascularization especially in combination with demographical (age above 62 years) and echocardiographical parameters (preoperative left ventricle ejection fraction below 50% and postoperative below 45%, respectively and left ventricle diameter postoperative diameter above 48 mm). Multivariable analysis revealed preoperative values of MLR > 0.2 and postoperative values of NLR > 3.5 as simple, reliable factors which may be applied into clinical practice for meticulous postoperative monitoring of patients with higher risk of worse prognosis.

Small remarks:

  1. Key words are given as abbreviations, maybe it is better to decipher them?

            Dear Reviewer, thank you for your valuable suggestion. It was corrected.

  1. PLR (median (Q1-Q3)) in table 1 are shown in a different form compared to the rest of the data in this table. Why? Table 2 shows a similar picture.

Dear Reviewer, thank you for your valuable suggestion. PLRs were within standard normal distribution, so the mean values and standard deviations were presented.

  1. Table 3 shows the value of 22.51627 for AUC, is this a typo? The value ranges from 0 to 1.

            Dear Reviewer, thank you for your valuable suggestion. It was corrected.

  1. Many citation errors [Error! Bookmark not defined.].

            Dear Reviewer, thank you for your valuable suggestion. It was corrected.

             Kind regards

       Tomasz Urbanowicz

Reviewer 2 Report

Dear Editor and authors, I have read the paper with great interest. Authors performed extensive exploratory overview of complete blood count information pre and post operatively in a large sample of CABG patients. I have following comments: 

1) in the title and the abstract the abbrevation OPCABG is not explained

2) there is missing P value for preoperative MCHC comparison between survivors and nonsurvivors

3) is there a typo for </> sign and MCHC who was shown to be protective if elevated and HR below one when analyzed as a continuous variable but above one when dichotomized with >21.1.

4) why was RDW that is the most potent inflammatory biomarker from the complete blood count not analyzed in the current context and other parameters like MCHC were. RDW should be readily available and certainly belongs to this context.

5) There are formating errors in the discussion section regarding references and the conclusion section is left unchanged with generic statement.

Author Response

Poznan, December 18th, 2021

Dear Reviewer,

Thank you for your valuable comments. We corrected the manuscript according to them.

As follows:

1) in the title and the abstract the abbrevation OPCABG is not explained

Dear Reviewer, thank you for your valuable suggestion. It was corrected.

2) there is missing P value for preoperative MCHC comparison between survivors and nonsurvivors

Dear Reviewer, thank you for your valuable suggestion. It was corrected.

3) is there a typo for </> sign and MCHC who was shown to be protective if elevated and HR below one when analyzed as a continuous variable but above one when dichotomized with >21.1.

Thank you for you suggestion. We changed the sign into above 21.1 to continue keep HR in the same direction.

The multifactorial model was presented including MCHC as added to the manuscript.

4) why was RDW that is the most potent inflammatory biomarker from the complete blood count not analyzed in the current context and other parameters like MCHC were. RDW should be readily available and certainly belongs to this context.

Dear Reviewer, thank you for your valuable suggestion. The analysis was based on most frequently applied inflammatory markers as NLR, MLR, PLR which are related to lymphocytes. We focused on ratios that are related to lymphocyte counts. We did not include RDW in our analysis, but we would like to thank you for the suggestion. We will proceed with your suggestion in the future, and include this marker in forthcoming analyses, as well.

5) There are formating errors in the discussion section regarding references and the conclusion section is left unchanged with generic statement.

Dear Reviewer, thank you for your valuable suggestion. It was corrected.

The following conclusion were added:

Hematological indices, NLR and MLR can be regarded as significant predictors of all-cause long-term mortality after off – pump surgical revascularization especially in combination with demographical (age above 62 years) and echocardiographic parameters (preoperative left ventricle ejection fraction below 50% and postoperative below 45%, respectively and left ventricle diameter postoperative diameter above 48 mm). Multivariable analysis revealed preoperative values of MLR > 0.2 and postoperative values of NLR > 3.5 as simple, reliable factors which may be applied into clinical practice for meticulous postoperative monitoring of patients with higher risk of worse prognosis.

        Kind regards

   Tomasz Urbanowicz

Round 2

Reviewer 1 Report

The authors responded to all the comments of the reviewer and significantly revised the manuscript. In its present form, the article can be recommended for publication. 

Reviewer 2 Report

Thank you, im that cade you should clearly state that rdw was deliberately ommited from analyses.